# Structure of the membrane-bound formate hydrogenlyase complex from *Escherichia coli*

Ralf Steinhilper [1], Gabriele Höff[2], Johann Heider[2,3] & Bonnie J. Murphy [1] ✉

The prototypical hydrogen-producing enzyme, the membrane-bound formate hydrogenlyase (FHL) complex from *Escherichia coli*, links formate oxidation at a molybdopterin-containing formate dehydrogenase to proton reduction at a [NiFe] hydrogenase. It is of intense interest due to its ability to efficiently produce $H_2$ during fermentation, its reversibility, allowing $H_2$-dependent $CO_2$ reduction, and its evolutionary link to respiratory complex I. FHL has been studied for over a century, but its atomic structure remains unknown. Here we report cryo-EM structures of FHL in its aerobically and anaerobically isolated forms at resolutions reaching 2.6 Å. This includes well-resolved density for conserved loops linking the soluble and membrane arms believed to be essential in coupling enzymatic turnover to ion translocation across the membrane in the complex I superfamily. We evaluate possible structural determinants of the bias toward hydrogen production over its oxidation and describe an unpredicted metal-binding site near the interface of FdhF and HycF subunits that may play a role in redox-dependent regulation of FdhF interaction with the complex.

When growing anaerobically, *Escherichia coli* performs mixed-acid fermentation yielding ethanol and organic acids[1]. Formate, the byproduct of fermentative conversion of pyruvate to acetyl-CoA, is initially exported by the FocA channel[2]. As the pH of the growth medium drops, formate re-enters the cell and synthesis of the formate-hydrogen lyase (FHL) complex is induced[3]. This heptameric membrane-bound complex catalyzes the disproportionation of formate to $CO_2$ and $H_2$ allowing efficient production of hydrogen from biomass[4], a process that was first described over 100 years ago[5]. Under standard conditions the disproportionation reaction is almost energy-neutral, but the driving force increases at low pH and low partial pressure of $H_2$[6], allowing it to drive ion translocation as demonstrated in *Thermococcus onnurineus*[7,8] and *Desulfovibrio vulgaris*[9]. More recently, studies showed that FHL can also work in reverse, catalyzing the $H_2$-dependent reduction of $CO_2$[10–12]. In spite of intense research, the complex eluded intact isolation and in vitro characterization until relatively recently. The [NiFe] hydrogenase of FHL, termed Hydrogenase 3 (Hyd-3), is one of four

hydrogenases encoded by the *E. coli* genome, but accounts for nearly all hydrogen production in vivo[13]. Hyd-3 is much more weakly product inhibited, and is more tolerant to CO, than other [NiFe] hydrogenases[6], making it well suited to biotechnological applications.

The group 4 membrane-bound hydrogenases (MBHs)[14], of which FHL is a member, share an evolutionary link to respiratory complex I (NADH:ubiquinone oxidoreductase)[15,16] and group 4 MBHs have been shown to couple turnover with proton translocation across the membrane[17,18]. An energy-conserving character of FHL is suggested by sequence homology[19], and due to the fact that the ATP synthase inhibitor DCCD substantially reduces gas production by *E. coli*, suggesting a link of FHL to the membrane potential[20]. Studies using inverted membrane vesicles derived from anaerobically grown *E. coli* reported formate-dependent formation of a transmembrane proton gradient that was eliminated by knockout of *hyc* genes[21], although more recent studies have variably supported or questioned an energy-conserving character for FHL[10,12,22].

[1]Redox and Metalloprotein Research Group, Max Planck Institute of Biophysics, 60438 Frankfurt am Main, Germany. [2]Department of Biology, Laboratory for Microbial Biochemistry, Philipps University Marburg, 35043 Marburg, Germany. [3]Synmikro-Center for Synthetic Microbiology, Philipps University Marburg, 35043 Marburg, Germany. ✉e-mail: bonnie.murphy@biophys.mpg.de

The conserved architecture of the complex I superfamily consists of a mostly soluble arm catalyzing both redox half-reactions and a membrane-bound arm, through which protons can be pumped. This architecture appears to have emerged by interaction of a soluble [NiFe] hydrogenase with a multiple resistance and pH (MRP) sodium-proton antiporter[15,19,23]. In the course of evolution, the pocket originally containing the [NiFe] hydrogenase active site has been modified to bind and reduce quinone[24] in complex I, and turnover at this site is widely believed to be coupled to ion translocation across the membrane[25,26]. Surprisingly, recent structural studies of the archaeal *Pyrococcus furiosus* MBH[27] and related membrane-bound sulfane sulfur reductase (MBS)[28] complexes show the majority of the membrane arm rotated by 180° in the plane of the membrane with respect to the redox modules, as compared to complex I architecture. In all structures including MBH and MBS, the soluble arm exhibits a conserved association with a membrane-anchor subunit homologous to complex I subunit ND1 (see Supplementary Table 3), whose transmembrane helices (TMH)2-6 closely resemble half of an antiporter-like subunit in fold[25]. Until now a structural understanding of FHL has been lacking, impeding further insight into its role in bacterial bioenergetics, its bias toward hydrogen production and its links to other members of the complex I superfamily.

Here, we present full atomic models of the complex in the oxidized and reduced states, providing important insight into the function and regulation of FHL and related complexes.

## Results

### Aerobic and anaerobic purification of FHL and single-particle cryo-EM studies

The FHL complex was purified from a strain bearing an internal 10x His affinity tag in HycE[6], according to a modified protocol (see Methods). The structure of aerobically isolated FHL, determined to 3.4 Å resolution (focused refinements 3.0–3.4 Å) by cryo-EM single-particle analysis provided important detail on the organization of the heptameric complex (Fig. 1a, b). However, the map density for the hydrogenase [NiFe] active site and proximal iron-sulfur cluster G1 was weak and poorly resolved (Fig. 2a), which we attribute to oxidative damage. We developed an anaerobic isolation protocol that allowed us to purify a complex exhibiting 20-fold higher hydrogen-uptake activity $(6201 \pm 1445 \, s^{-1})$ compared to aerobically purified complex from this study $(253 \pm 164 \, s^{-1})$ (Fig. 2b) and previous studies $(2.44 \pm 0.62 \, U \, mg^{-1} \equiv 13 \, s^{-1})$[10]. The size exclusion chromatography (SEC) elution profile of the aerobically isolated complex shows two major peaks, the slower-eluting of which corresponds to a soluble complex lacking the membrane-associated subunits HycC and HycD (Supplementary Fig. 1a). In contrast, the SEC profile of anaerobically purified protein shows a single predominant peak, with a much smaller peak corresponding to the complex without HycC/HycD (Supplementary Fig. 2a). The formate dehydrogenase subunit FdhF, which shows partial occupancy in the aerobically purified complex, was entirely absent in the anaerobically isolated complex in multiple replicates of the purification. The structure of the anaerobically isolated complex was determined at an average resolution of 2.6 Å (Supplementary Fig. 2d, e). Together, these maps allowed us to build atomic models including sidechains for 94% of the aerobically isolated complex (heptameric FdhF-HycBCDEFG) and 97% of the anaerobically isolated complex (hexameric HycBCDEFG). The 2.6 Å resolution map of anaerobically isolated FHL also allowed us to model well-ordered lipids and water molecules.

In line with its higher hydrogen-uptake activity, the map derived from anaerobically prepared sample had stronger and higher-resolution density for the nickel-iron active site in HycE and the proximal iron-sulfur cluster G1 in HycG (Fig. 2a). Differences between the maps were also present in conserved loops at the interface region

between soluble and membrane arms, discussed in detail below. We suggest that oxidative damage to the [NiFe] site causes disorder in this region, leading to the partial dissociation of the membrane arm observed by SEC.

For the remainder of the results and discussion, except regarding the FdhF subunit and its attachment to the complex, we refer to the higher-resolution anaerobic model.

### Overall architecture of the FHL complex

The FHL complex exhibits the characteristic L-shaped structure of the complex I superfamily with membrane-embedded and soluble arms (Fig. 1a, d). The redox catalytic core is located entirely in the soluble arm of the complex (Fig. 1b). Formate is oxidized to $CO_2$ at a selenocysteine-containing molybdenum bis-molybdopterin guanine dinucleotide $(Mo(MGD)_2)$ active site within the formate dehydrogenase (FdhF). Our cryo-EM map obtained from aerobically prepared sample shows a $Mo(MGD)_2$ geometry similar to the crystal structure of oxidized FdhF in the [Mo(VI)]-state[29,30] (Fig. 3a, b). An electron relay containing eight [4Fe4S] clusters, coordinated by FdhF (A1), HycB (B1-B4), HycF (F1, F2) and HycG (G1), leads from the molybdopterin site to a [NiFe] hydrogenase active site in HycE (Fig. 1c). The inter-cluster distance along the relay is consistently shorter than 14 Å, allowing efficient electron transfer[31].

The membrane arm consists of the integral membrane proteins HycD and HycC, comprising a total of 24 TMHs (Fig. 6a). HycD mediates interaction of the membrane-bound antiporter-like subunit HycC with the soluble arm.

FdhF is loosely associated with the complex and previous studies reported substoichiometric amounts of Se and Mo, which was attributed to partial loss of the subunit during purification[6]. In our preparations, the subunit was lost partially in aerobically purified and completely in anaerobically purified sample. FdhF interacts with HycB via hydrophobic interactions and hydrogen bonds (H-bonds) (Fig. 4a and Supplementary Fig. 5) and with HycF electrostatically (Fig. 4a, b). Our data suggest that FdhF is more strongly associated with the Hyc subunits under oxidizing than under reducing conditions. This may be related to a regulatory process, although we cannot conclusively rule out the possibility that other difficult-to-control differences between aerobic and anaerobic protocols could lead to the differing observations. Previous studies have shown that FdhF may also interact with other proteins in the cell[32,33].

### Evolutionary relationship of FdhF and complex I subunit NDUFS1

According to a theory of modular evolution, complex I evolved from the association of a soluble [NiFe] hydrogenase with an Mrp-type sodium-proton antiporter[15,19,23]. An early form of this respiratory system is represented by the energy-conserving group 4 membrane-bound hydrogenase (MBH) from the hyperthermophilic archaeon *P. furiosus*, a complex that accepts electrons from oxidative reactions in the cell via the electron transfer protein ferredoxin[17]. In contrast, like complex I, FHL catalyzes both the oxidative and reductive half-reactions directly. Molybdopterin-containing enzymes including FdhF exhibit sequence-[34] (Supplementary Fig. 8), and structural[35] homology to the complex I subunit NDUFS1 (Supplementary Table 3). Sequence comparison suggests that NDUFS1 forms part of an alternate electron-input module, which may be required in bacterial 12-subunit complex I analogs as they lack an NADH-oxidizing module[36]. In other bacterial species, the pathway appears to have become a relic, leaving the NDUFS1-ligated [4Fe4S] cluster N7 too remote to play a role in electron transfer between active sites[37]. Sequence similarity of FdhF and HycB to NDUFS1[19] raises the possibility that an FHL-type complex may be the most recent MBH ancestor of complex I.

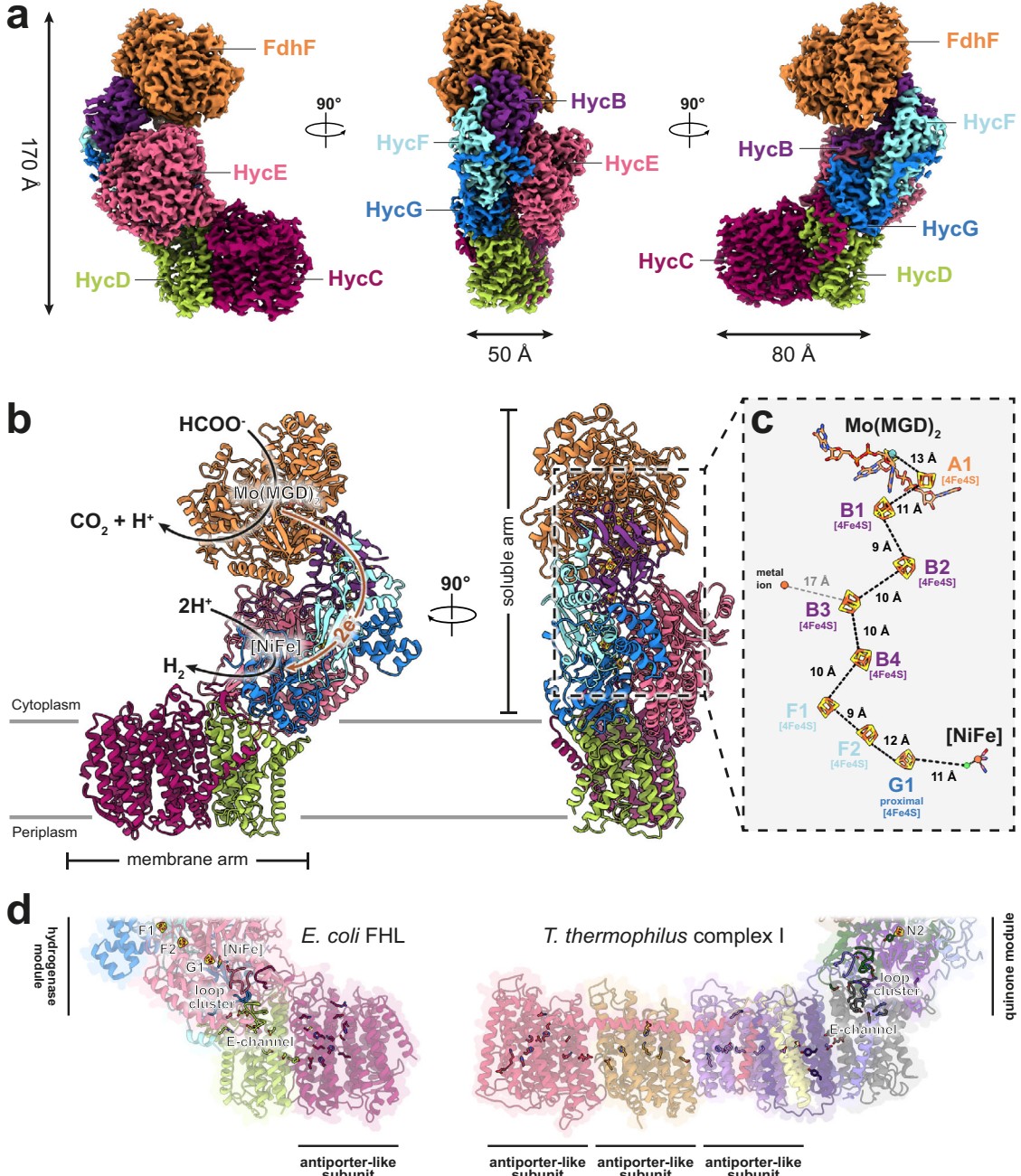

**Fig. 1 | Overall architecture and atomic model of *E. coli* FHL. a** Composite cryo-EM map of aerobically isolated FHL segmented and colored by subunit. Except where otherwise indicated, subunit coloring is consistent throughout the manuscript. **b** Atomic model of aerobically isolated FHL. Formate oxidation occurs at the $Mo(MGD)_2$ site in FdhF and electrons are transferred to the [NiFe] hydrogenase HycE, driving proton reduction to $H_2$. The soluble arm is connected to the membrane arm via subunit HycD. **c** An iron-sulfur cluster relay with distances less than 14 Å connects the active sites. The [4Fe4S] clusters are labeled and colored according to their subunit. A metal ion in subunit HycF is 17 Å from the [4Fe4S] cluster B3 in HycB. **d** Comparison of *E. coli* FHL and *Thermus thermophilus* complex I (PDB 4HEA[25]) architecture. The hydrogenase module of FHL is homologous to the quinone reductase module of complex I. Charged residues connect the FHL [NiFe] active site with the membrane arm, similar to the E-channel of complex I. Select hydrophilic residues in the antiporter-like subunits of both complexes are shown. Complexes in **d** are positioned such that antiporter-like subunits (HycC and ND2/4/5) are in the same orientation.

Superposing our structure with the C-terminal region of *Thermus thermophilus* Nqo3 (NDUFS1) gives an overlap of the [4Fe4S] cluster A1 of FdhF with the off-pathway [4Fe4S] cluster N7 of Nqo3 (NDUFS1) (Supplementary Fig. 6a). However, the way in which FdhF interfaces with the FHL complex is different to that seen for Nqo3 (NDUFS1) in complex I (Supplementary Fig. 6b, c). The significant difference in architecture suggests that in spite of the sequence homology between these subunits, their association to members of the complex I superfamily is likely to have arisen from independent evolutionary events.

A recently published structure of the soluble *Rhodobacter capsulatus* FDH complex[38] revealed a subunit architecture and cofactor arrangement much more similar to that seen in Nqo3 (NDUFS1), in which the equivalent of the off-pathway cluster N7 connects to the electron relay via an additional [4Fe4S] cluster. In contrast, the way in which FdhF interacts with HycB is more similar to the interaction of the α- and β-subunits of the *E. coli* formate dehydrogenase-N[39], a homolog of FdhF that plays a role in nitrate respiration (Supplementary Fig. 6d).

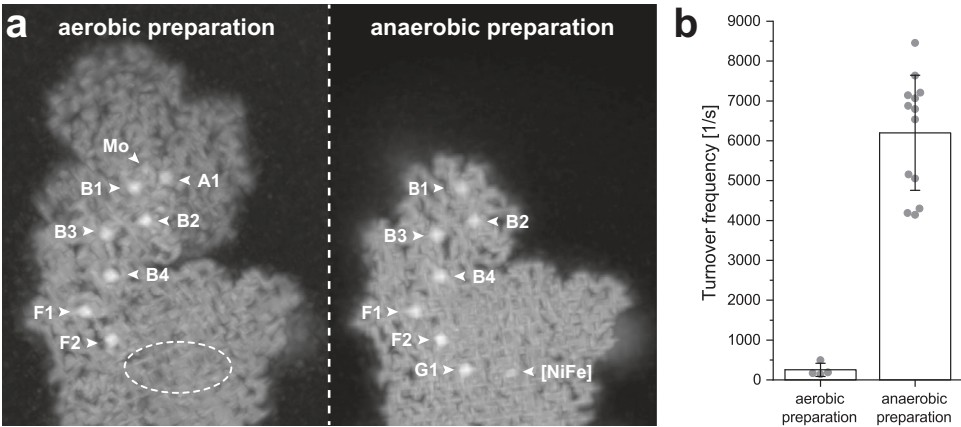

**Fig. 2 | Comparison of aerobically and anaerobically prepared FHL. a** Solid-view representation of unsharpened maps of aerobically prepared FHL shows strong intensity for the [4Fe4S] clusters in FdhF (A1), HycB (B1-B4) and HycF (F1, F2). Density for the proximal [4Fe4S] cluster G1 in HycG and the [NiFe] cofactor in HycE is weak (the approximate position is indicated with a dashed ellipse). The anaerobically prepared FHL shows strong density for the proximal [4Fe4S] cluster (G1) and the [NiFe] cofactor. **b** Hydrogen-uptake activity of aerobically ($253 \pm 164\,\mathrm{s^{-1}}$) and anaerobically ($6201 \pm 1445\,\mathrm{s^{-1}}$) prepared FHL. Values indicate mean $\pm$ standard deviation. Initial slopes were determined for 4 independent assays from $n = 1$ biologically independent samples for the aerobic preparation and from 13 independent experiments from $n = 2$ biologically independent samples for the anaerobic preparation. Source data are provided as a Source Data file.

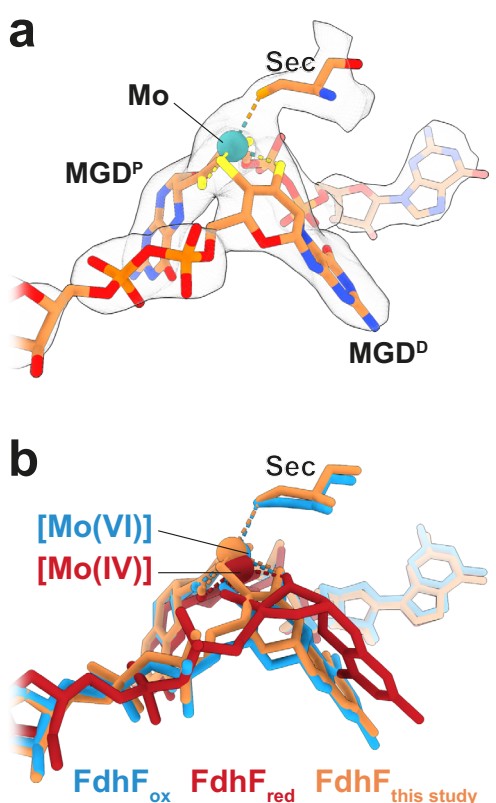

**Fig. 3 | Geometry of the Mo(MGD)$_2$ cofactor. a** Atomic model and cryo-EM density of the Mo(MGD)$_2$ cofactor with the proximal MGD (MGD$^P$), distal MGD (MGD$^D$), and the metal-coordinating selenocysteine (Sec) in subunit FdhF of the aerobically isolated FHL. **b** Superposition of our aerobically isolated FHL model with crystal structures of oxidized (blue; PDB 1FDO[29]) and reduced (red; PDB 2IV2[30]) FdhF. The geometry of MGD$^D$ is similar to the crystal structure of FdhF in the oxidized [Mo(VI)] state.

## FdhF interacts with HycB and an unpredicted metal ion of HycF

Electron transfer between the electron-input module (FdhF) and the primary hydrogenase modules (HycG and HycE) is mediated by the ferredoxin-like proteins HycB and HycF, as predicted from genetic experiments[40]. A loop region between the B3 and B2 [4Fe4S]-cluster-ligating cysteine motifs of HycB interacts extensively with HycF and HycE. HycF has an N-terminal extension that comprises a long loop, with a short amphipathic helix near the membrane surface at the extreme N-terminus. This structural feature is present in all structures of the complex I superfamily solved to date and has been shown to bind cardiolipin[41], with a suggested influence upon global structural dynamics and quinone access to the active site[42]. The C-terminal extension of HycF interacts with HycB and contains a cysteine quartet (Cys117, Cys120, Cys155 and Cys158) that coordinates an unpredicted cofactor at the periphery of the protein, consistent with a single metal ion that we have tentatively assigned as iron. This is consistent with previous inductively coupled plasma mass spectrometry (ICP-MS) analysis that found a higher Fe:Ni ratio than expected given the predicted cofactors, and that did not report additional metals[6]. The coordination sequence (CxxC-x$_{34}$-CPxC) bears similarity to ligand-binding sequences for rubredoxin-type Fe$^{2+/3+}$ centers, or Zn$^{2+}$. Neither the fold nor the detailed sequence closely matches characterized rubredoxins (Supplementary Fig. 10b), though we note some similarity in secondary structure around the first CxxC motif. The metal ion is about 17 Å from the B3 [4Fe4S] cluster of HycB (Fig. 1c). The position of the ion argues against a role in conducting electrons between active sites, however the ion could be in redox equilibrium with the iron-sulfur cluster relay: Marcus theory predicts electron transfer on the order of $10^2\,\mathrm{s^{-1}}$ over a distance of 17 Å for $\Delta G = 0$[31]. The metal-binding site forms part of a positively charged patch of HycF that is directly opposite a patch of acidic residues of FdhF (Fig. 4a). We suggest that a redox-active ion in this binding site could play a role in regulating affinity of the complex for FdhF as reduction of the ion would decrease the strength of charge-charge interaction between the two sites. Such a mechanism would be expected to lower the rate of hydrogen-dependent CO$_2$ reduction by the complex, since this reaction requires interaction of a reduced Hyc complex with FdhF. Arg122 of HycF, which is near to the ion, exhibits two alternate conformations in the aerobically isolated structure: one facing toward the ion and the other toward the acidic patch of FdhF (Fig. 4b). The structure of anaerobically isolated FHL shows the arginine in a single conformation facing the metal ion (Fig. 4b). The ion-binding motif and Arg122 are conserved in FHL complexes from several other bacteria and the HyfH subunit of the related *E. coli* FHL-2 complex but not in the homologous complex I subunit NDUFS8 (Supplementary Fig. 10a).

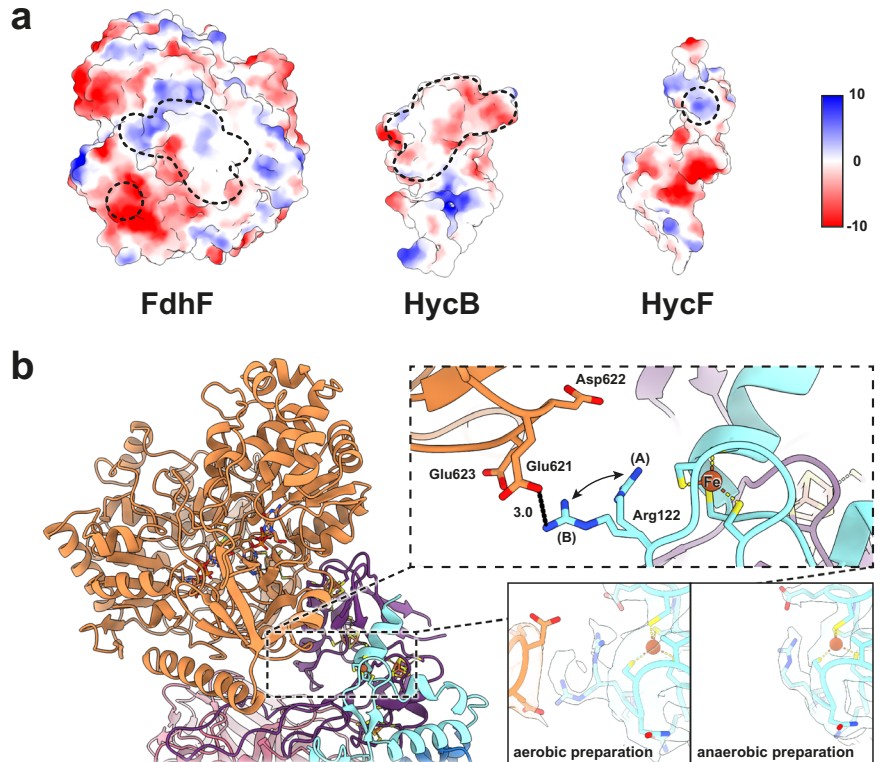

**Fig. 4 | The formate dehydrogenase H (FdhF) interacts with HycB and HycF.**
**a** Electrostatic potential representation of FdhF, HycB, and HycF; the interface between the subunits is outlined with dashed lines. HycB and FdhF interact via large hydrophobic patches and H-bonds (Supplementary Fig. 5). Electrostatic interactions occur between a small electropositive region of HycF and an electronegative patch of FdhF. **b** Detailed view of the interaction site between FdhF and HycF. Arg122 of HycF, which sits near to an unpredicted metal ion of HycF and within a small electropositive region, shows two alternative sidechain conformations in the map obtained from aerobically prepared sample. In conformation A, it is close to the putative iron, while in conformation B, it forms a salt bridge with Glu621 of a negatively charged patch on FdhF. In the map derived from anaerobically prepared sample, Arg122 is only present in conformation A.

## Mapping hydrogenase catalysis onto the complex I superfamily

The minimal architecture of [NiFe] hydrogenases comprises a small subunit harboring up to three iron-sulfur clusters and a large hydrogenase subunit coordinating the [NiFe] active site[43]. HycE, the large hydrogenase subunit of FHL, is a fusion protein, with the N-terminal residues 1–151 bearing homology to complex I NDUFS3, while residues 182-537 are homologous to the large subunit of soluble hydrogenases and the quinone-binding subunit NDUFS2 of complex I[44] (Supplementary Table 3). The C-terminal portion of HycE coordinates the [NiFe] cofactor. Our experimental density is consistent with a standard [(Cys-S)$_2$-Ni-(μ$_2$-Cys-S)$_2$-Fe(CN)$_2$CO] catalytic cluster. This site has evolved into a quinone-reducing pocket in complex I. The small hydrogenase subunit, HycG, is homologous to NDUFS7 of complex I. HycG ligates the proximal [4Fe4S] cluster G1 in a position equivalent to cluster N2 of complex I, but the consecutive coordinating cysteine residues and the redox-Bohr histidine typical of complex I cluster N2[35] are absent in MBHs (Supplementary Fig. 11 and Supplementary Fig. 12).

The catalytic mechanism of [NiFe] hydrogenases has been most extensively studied for soluble hydrogenases, and is summarized in recent reviews[45,46]. Similarity in sequence and structure support a conserved heterolytic mechanism for the reversible interconversion of H$_2$/2H$^+$ across different types of [NiFe] hydrogenases.

The active site is buried deep within the polypeptide, so that protons, electrons and hydrogen require pathways to exchange with the environment[47]. The electron pathway provided by the iron-sulfur clusters is well established[43] and molecular hydrogen appears to be able to enter or leave the active site via several hydrophobic routes as shown by MD simulations and crystallographic studies[48]. Highly adapted proton transfers are central to efficiency of electrocatalysts including hydrogenases[46] and several studies have traced the pathway taken by protons to/from the [NiFe] active site. At least one of the protons is transferred to a terminal cysteine ligand of nickel, (Cys546 in *D. vulgaris* HydB; Cys531 in *E. coli* FHL), and from there to a strictly conserved carboxylate (Glu34 in *D. vulgaris* HydB) that is essential for function[49,50] and occupies a position equivalent to a highly conserved histidine (His95 in *Y. lipolytica* NDUFS2) of complex I (Fig. 5a, d) widely believed to donate one substrate proton for quinone reduction[51,52]. These proton-donating Glu/His residues are located on a conserved loop region between two beta sheets, known as the β1-β2 loop. MBH sequences generally contain two adjacent acidic residues at this position (Asp192 and Glu193 in *E. coli* HycE, Fig. 5a, d). Our structure shows Glu193 closer to Cys531-S suggesting that this residue exchanges protons with Cys531, in spite of the fact that a sequence alignment of soluble and membrane-bound hydrogenases shows Asp192 of HycE at the equivalent position to Glu34 of *D.* vulgaris HydB (Fig. 5d). This suggestion is further supported by the fact that *Methanosarcina barkeri* EchE contains an acidic residue equivalent to HycE Glu193, but not to HycE Asp192. *P. furiosus* MbsL also contains a single acidic residue on the β1-β2 loop, although Yu et al.[28] have suggested that substrates are reduced but not protonated in the MBS active site pocket.

The β1-β2 loop appears less restricted in FHL than in soluble hydrogenases, having fewer stabilizing H-bonds (Fig. 5c). In FHL the Glu193 is too far (4 Å) from Cys531-S to allow direct proton transfer (Fig. 5a), suggesting that the β1-β2 loop is likely flexible during turnover. Loop flexibility has been observed in complex I[53] and was suggested to occur in *P. furiosus* MBH on the basis of MD simulations[54].

From the β1-β2 loop glutamate, several proton pathways have been proposed, based on MD simulations[55] and high-resolution X-ray

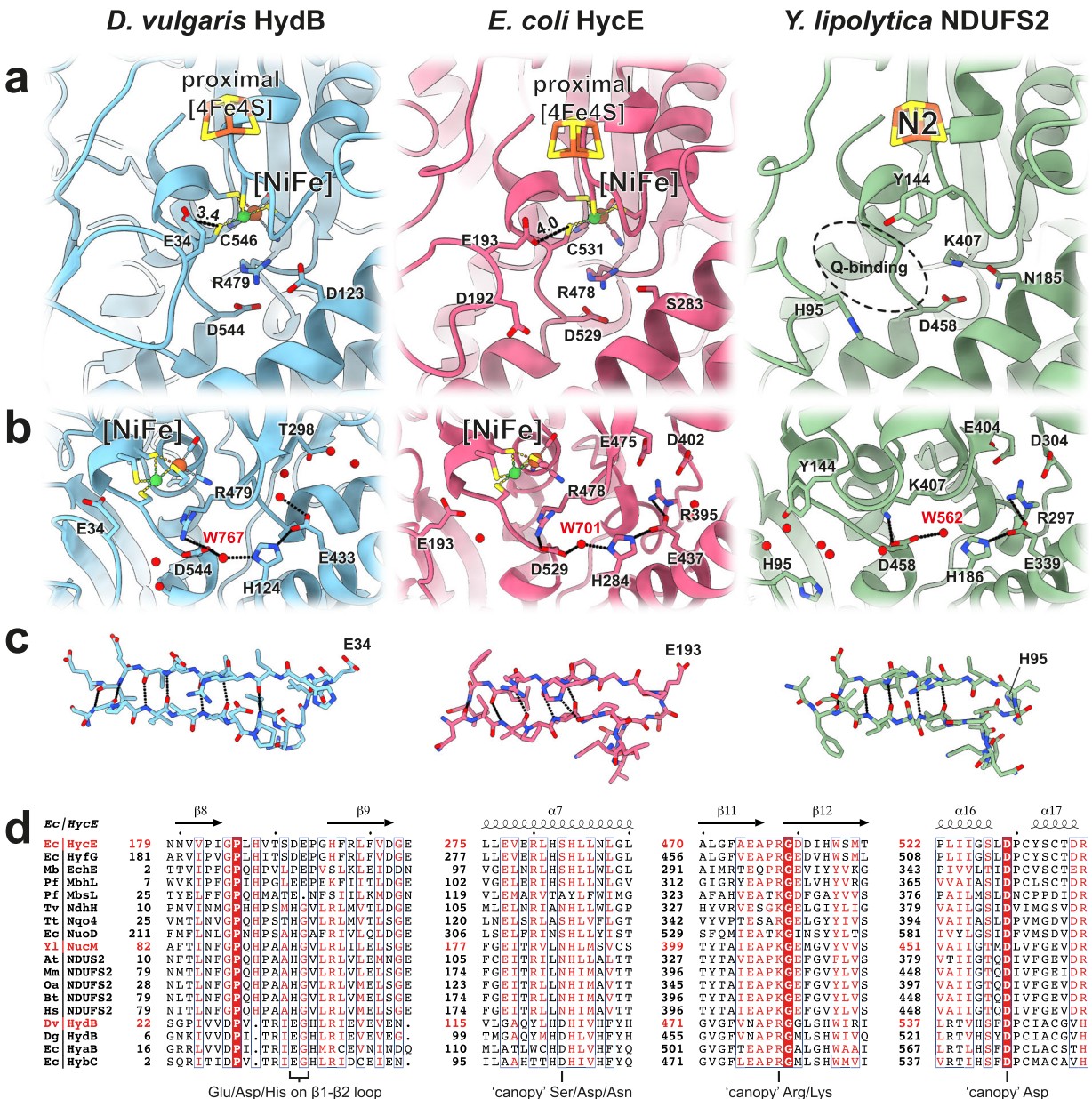

**Fig. 5 | Catalytic sites and substrate proton pathways in soluble [NiFe] hydrogenases, FHL and complex I. a** Comparison of putative proton-transferring residues in a soluble [NiFe] hydrogenase (PDB 4U9H[56]), FHL and complex I (PDB 7O71[41]). Distances between the conserved β1-β2 loop glutamate (HydB E34, HycE E193) and the terminal cysteine (HydB C546, HycE C531) are indicated in Å. In complex I the β1-β2 loop harbors a conserved histidine (NDUFS2 H95), which is believed to donate a proton for quinone reduction. The canopy residues (HydB R479/D544/D123, HycE R478/D529/S283, NDUFS2 K407/D458/N185) are located in proximity to the active site of the enzymes. **b** Proton pathways extend from the active site to the bulk via conserved residues (HydB R479/D544/H124/E433, HycE R478/D529/H284/E437/R395, NDUFS2 K407/D458/H186/E339/R297). Select water molecules in HydB, HycE and NDUFS2 are shown as red spheres; select hydrogen bonds are marked with dashed lines. **c** The β1-β2 loop region of HydB appears to be more rigid than those of HycE and NDUFS2, which form fewer H-bonds (dashed lines) between the beta strands. **d** Conservation of β1-β2 loop and canopy residues in the complex I superfamily and [NiFe] hydrogenases. The position of the proton-donating residues (HydB E34, HycE D192/E193, NDUFS2 H95) are indicated. The canopy residue HycE Ser283 is conserved as a serine, aspartate or asparagine (except in MbsL). HycE Arg478 is a conserved arginine in hydrogenases, which is substituted by a lysine in most representatives of complex I. HycE Asp529 is strictly conserved.

crystallography structures[56] of soluble [NiFe] hydrogenases, but the residues involved in many of these pathways are not strictly conserved in MBHs (Supplementary Fig. 7). Mutational evidence[57,58] and comparison to [FeFe] hydrogenases and synthetic catalysts suggests a role for the strictly conserved canopy residues (HycE Arg478/Asp529) in the first proton abstraction from $H_2$ (or, in the direction of $H_2$ production, protonation of a metal-bound hydride). The canopy residues lie at the end of a putative proton transfer pathway including HycE His284, Glu437 and Arg395, which appears to be conserved in FHL,

MBH[27], soluble hydrogenases[56] as well as in complex I[59] (Fig. 5b). The canopy arginine (HycE Arg478) is present as either an arginine or more commonly a lysine in complex I, while the canopy aspartate (HycE Asp529) appears to be strictly conserved (Fig. 5d). Mutation of either residue to non-protonatable amino acids substantially impairs quinone reductase activity of complex I[60]. Studies of complex I have suggested the β1-β2 loop may be reprotonated through the TMH1-2 loop of subunit ND3[61] (HycC homolog, Supplementary Table 3) or by the so-called E-channel[25], a glutamate-rich pathway connecting the quinone

reduction site to the membrane arm (Fig. 1d), although the proton pathway to the E-channel remains under debate[41,53].

## Structural basis of hydrogen production

Hyd-3, the hydrogenase of the FHL complex, exhibits a bias toward hydrogen production and a very low degree of product ($H_2$) inhibition for $H_2$ production, more similar to the unrelated [FeFe] hydrogenases than to other [NiFe] hydrogenases[6]. The structural basis of this difference is unknown and is of critical relevance to the design of biological and synthetic catalysts. Our structures suggest two possibilities for this observed difference. Ser283 of HycE (Fig. 5a), one of three so-called canopy residues at the [NiFe] site, is variably an Asp, Asn, or Ser in other [NiFe] hydrogenases (Fig. 5d). This position is frequently occupied by Asp in $H_2$-oxidizing enzymes, while both the MBHs and the relatively distantly related [NiFeSe] hydrogenases contain a Ser at this position[62], and both groups exhibit enhanced $H_2$ production activity[6,63]. Mutation of this residue (HyaB D118) to alanine in the $H_2$-oxidizing hydrogenase-1 (Hyd-1) from *E. coli* resulted in an enzyme exhibiting an extremely high $K_M(H_2)$; however, this mutant did not produce $H_2$[58,62]. As noted by the authors, this could be due to other limitations on $H_2$ production in Hyd-1[64]; a mutation of D118 to serine was not reported to our knowledge. A second possibility is that the conformational flexibility of the β1-β2 loop may play a role in tuning the properties of the active site. Recent MD simulations of the *P. furiosus* MBH showed that the β1-β2 loop comes nearer to the [NiFe] site in its protonated state than in its deprotonated state[54]. Such a dependence would favor the protonated state of the [NiFe] cluster, and may thereby disfavor binding of $H_2$. In this way, the enhanced flexibility of the β1-β2 loop could underlie a tendency toward $H_2$ production, even in the presence of high ρ($H_2$).

## Conserved loops form the interface between catalytic and membrane subunits

The β1-β2 loop lies at the interface of the soluble and membrane arms along with three additional peptide loops, which are proposed to play a key role in coupling turnover with proton translocation across the membrane in complex I[53]; all four loops have homologs in FHL (Supplementary Fig. 3). For simplicity, the complex I nomenclature for these loops will be used throughout. The ND3 loop (HycC TMH15-THM16 loop), β1-β2 loop (HycE β8-β9 loop), NDUFS7 loop (HycG residues 69-80) and the ND1 loop (HycD TMH5-TMH6 loop) (Supplementary Fig. 3b–e) are all well-resolved in our anaerobic cryo-EM map and could be modeled with confidence. In the map obtained from aerobically prepared sample only weak density was observed for the ND1 and β1-β2 loops (Supplementary Fig. 4a, d). The ND3 loop, which is frequently disordered in complex I structures and the structure of *P. furiosus* MBH, is in a similar conformation to that observed for active mouse complex I[65], with the conserved Cys537 of HycC in a binding pocket formed by Phe122 of HycD and His188 of the β1-β2 loop of HycE.

## The membrane arm harbors a putative substrate proton pathway

The membrane arm of complex I homologs includes the highly conserved membrane-anchor subunit (ND1 in complex I nomenclature) together with a variable number of antiporter-like subunits. FHL has a minimal membrane arm consisting only of HycD (ND1 homolog) and the antiporter-like subunit HycC (ND2, ND4, ND5 homolog; Supplementary Table 3). In contrast to *P. furiosus* MBH, which is believed to establish a sodium gradient[66], FHL lacks homologs to MbhA-G.

HycD has eight highly tilted transmembrane helices (Fig. 6a) that build up a hydrophilic cavity connecting the hydrogenase active site with the center of the membrane arm (Fig. 6c). The cavity is lined with a number of conserved charged residues, similar to other members of the complex I superfamily. Compared to complex I architecture, HycD

is rotated by 180° in the plane of the membrane relative to HycD in FHL (Fig. 1d), consistent with structures of the MBH and MBS complexes of *P. furiosus*. Unlike the *P. furiosus* MBH structure, in which the membrane subunits MbhM and MbhH are separated by lipids[27], HycC and HycD are tightly packed against each other, allowing close contact between residues of the proposed proton-conducting E-channel. HycC comprises 16 TMHs (Fig. 6a) exhibiting strong homology to antiporter-like subunits of complex I[25], MBH[27], MBS[28] and MRP[67], with characteristic broken helices in TMH7 and TMH12 that provide flexibility and are thought to play a role in proton pumping in the complex I superfamily.

In FHL the E-channel extends from Glu201, Glu206 and Glu199 on the conserved ND1 loop along Glu130, Glu189, Arg108, Glu138 to Tyr102, which is located at the subunit boundary approximately 4 Å from the adjacent Glu391 in HycC TMH12 (Fig. 6c). Site-directed mutagenesis of HycD shows that substitutions of residues along this pathway, reaching as far as Glu138 at the interface of HycD/HycC (Fig. 6c), strongly impact hydrogen production in vivo[10]. This provides strong support for the idea that either a substrate proton travels through this pathway to the [NiFe] active site, or that turnover is tightly coupled to proton translocation via this site. Interestingly, HycC Glu391 is only conserved for complex I subunit ND4 and its homologs, differing from other group 4 membrane-bound hydrogenases, in which a highly conserved lysine takes up this position (Supplementary Fig. 14a). Densities consistent with water suggest well-ordered water molecules form H-bonds with Glu391. We examined the structure for proton pathways from the solvent to Glu391 and found that FHL differs from previously characterized homologs in this regard. A path from the N-side conserved in complex I subunit ND5, MrpA, MbhH and MbsH/MbsH′, is interrupted in HycC by Gln285 and Ala230 (Fig. 6c and Supplementary Fig. 14a). Instead, a pathway including residues Glu294, His291, Cys430 and Lys433, which is not conserved in MBH, MBS or complex I, leads from Glu391 to the N-side. This observation is consistent with a previous mutational study showing that HycC mutants E391A and D354A reduced hydrogen production by 45%, whereas E135A, H222A, K239A, T292A, H328A and K336A mutations did not cause a significant decrease in hydrogen production in vivo[10]. It was previously noted that the lysine of a conserved Glu/Lys pair in antiporter-like subunits is substituted by Leu208 in HycC[10,16], while the strictly conserved Glu135 on TMH5 is present. These residues are located toward the membrane-exposed end of the HycC subunit. We did not observe a clear proton pathway connecting the E-channel to the P-side of the membrane.

While the majority of HycC (N-terminal residues 1-485) is homologous to antiporter-like subunits, the C-terminal extension (residues 500-608) is homologous to subunit ND3 (Supplementary Fig. 14b). This region includes two transmembrane helices, separated by a long loop, equivalent to the ND3 loop. A similar fusion is seen in MbsH′ of MBS[28]. A lateral helix (HL) between TMH15 and TMH16 is a conserved element of antiporter-like subunits in complex-I related enzymes. The helix length varies between different members of the superfamily to accommodate interaction with between one (FHL) and three antiporter-like subunits (complex I)[16]. Additional density between the helix HL and TMH7 of HycC could be confidently assigned as cardiolipin (CDL). Arg567, Tyr290, His215, Gln220 and Asn571 of HycC interact with the headgroup of CDL (Fig. 6b), whereas non-polar residues in the broken helix in TMH7 interact with the alkyl chains. Cardiolipin plays an important role in the stability and activity of respiratory complexes in prokaryotes[68] and eukaryotes[69] including complex I.

## Discussion

Our structures of the FHL complex reveal the basis of hydrogen production in *E. coli*. We show that anaerobic isolation results in vastly higher hydrogenase activity, and differences between the aerobic and anaerobic structures indicate that irreversible damage occurs at the [NiFe] site and proximal iron-sulfur cluster when the protein is purified

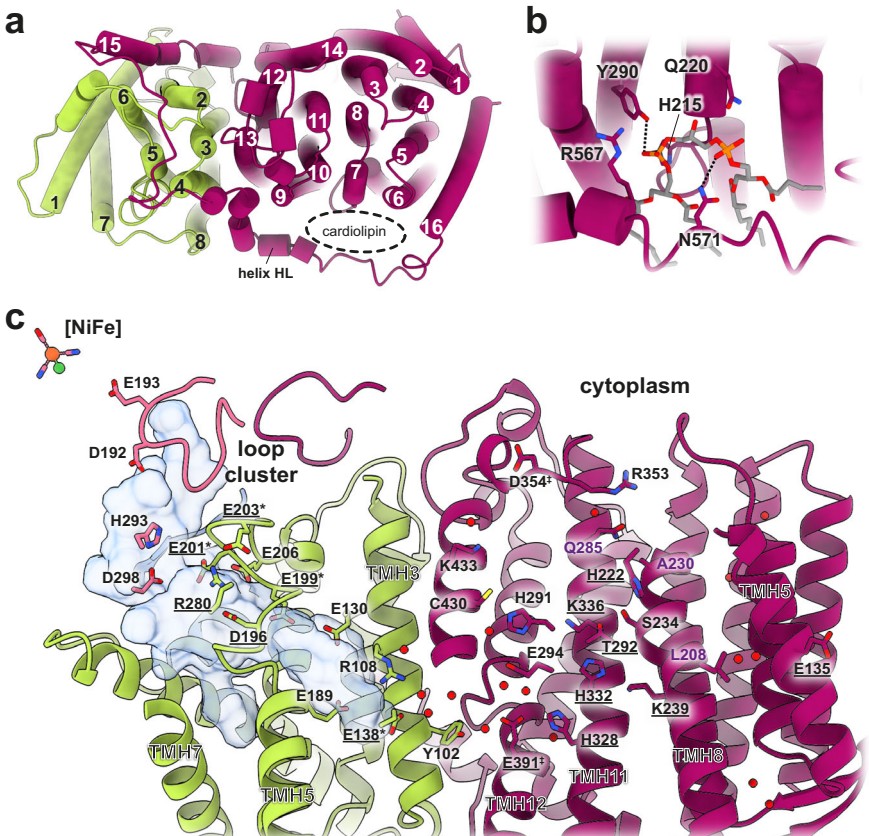

**Fig. 6 | Membrane subunits and putative substrate proton pathways. a** Top view of the FHL membrane arm. HycD contains eight transmembrane helices (TMHs), which are highly tilted. HycC is an antiporter-like subunit consisting of 16 TMHs with a short lateral helix (helix HL) between TMH15 and TMH16. Typical for antiporter-like subunits are the broken helices TMH7 and TMH12. A cardiolipin (CDL) molecule is clamped between the helix HL and the antiporter-like part of HycC. **b** The headgroup of CDL is stabilized by Arg567, Tyr290, His215, Gln220, and Asn571. Atoms in H-bond distance are marked with dashed lines. **c** Detailed view of the FHL membrane arm. HycD harbors a hydrophilic cavity (analyzed with CASTp 3.0[91] using a 1.4 Å probe) that connects the β1-β2 loop close to the [NiFe] active site with the membrane. Several hydrophilic residues line the E-channel (HycD Glu201, Glu206, Glu199, Glu130, Glu189, Arg108, Glu138, and Tyr102), residues with conservation in complex I are underlined. Tyr102 is about 4 Å from the adjacent, highly hydrated Glu391 of HycC. From there, a connection to the cytoplasm (N-side) is mediated by residues Glu294, His291, Cys430, Lys433. Conserved hydrophilic residues include His328, His332 and Lys336 on TMH11, as well as Lys239 on TMH8. Lys336 is connected to the conserved His222 via Thr292 and Ser234, but the connection to the N-side via a conserved Thr/Ser on TMH8 and Lys/Arg on TMH11 is not present in HycC (substituted by Ala230 and Gln285, respectively). The conserved Lys/Glu pair between TMH7 and TMH5 is replaced by a Leu208 on TMH7, but Glu135 on TMH5 is present. Select water molecules are shown as red spheres. *mutation of HycD residues E138 and E199/E201/E203 reduced in vivo $H_2$ production by 98% and 99%, respectively[10], ‡mutation of HycC residues E391 and D354 each reduced in vivo $H_2$ production by 45%[10]. Some parts of the model are not displayed for figure clarity.

aerobically. Our results suggest that the FdhF subunit is less stably attached to the complex under reducing conditions. This observation could be related to a redox-state change of an unpredicted ion coordinated by a cysteine quartet in HycF near to the interface with FdhF. Further biochemical and mutational studies will be required to evaluate this proposal.

Structural similarity to respiratory complex I is obvious for six of the seven subunits of our atomic model. In spite of sequence and structural homology of FdhF to Nqo3, the interaction of FdhF with the rest of the complex does not closely match that seen for Nqo3.

It is remarkable that, throughout evolution of the [NiFe] site to accommodate binding and reduction of a large hydrophobic substrate, the proton-donating role of the β1-β2 loop appears to have been conserved. The β1-β2 loop may be reprotonated via the E-channel, with substrate protons entering through the membrane arm. In support of this, our structure shows a clear pathway for protons from the cytoplasm to the E-channel via the antiporter-like subunit HycC, in spite of the fact that some residues of a previously described pathway are not conserved. Further studies will be required to understand the role that FHL plays in cellular bioenergetics.

Looking more broadly at conservation throughout the complex I superfamily, sequence and structural conservation of substrate proton pathways and of loops at the interface of the membrane and soluble arms, throughout several billions of years of evolution, strongly support the idea that a conserved core mechanism underlies coupling of redox chemistry to ion translocation across the superfamily. An obvious but critical difference between MBHs and complex I is that the product of the MBH reaction ($H_2$) is a small, non-polar gas molecule bearing no discernible similarity to quinol. Therefore, an evolutionary perspective speaks against any mechanistic proposal that fundamentally depends on quinone/quinol dynamics, favoring instead mechanisms that rely on movement of charges (electrons and/or substrate protons) through the complex.

## Methods
### Protein expression and purification
*E. coli* strain MG059e1[6], which harbors a sequence encoding a His$_{10}$-tag placed after Gly-83 in the *hycE* gene of the *hycABCDEFGHI* operon was expressed in LB media supplemented with 0.4% glucose by anaerobic fermentation in a 200 L reactor (Sartorius). The culture was inoculated

with 400 mL of aerobically grown preculture and stirred at 50 rpm for 16 h at 37 °C. Cells were harvested anaerobically in a flow centrifuge (Carl Padberg Zentrifugenbau), flash frozen in liquid $N_2$ and stored at −80 °C. Both the aerobically and anaerobically purified samples were prepared from these cells.

For the aerobic isolation, 10-20 g of frozen cell pellet was resuspended in lysis buffer (20 mM Tris, 200 mM NaCl, 1 mM DTT, pH 7.5) containing 50 μg/mL lysozyme, 10 μg/mL DNase I and protease inhibitor cocktail (Roche Diagnostics) and passed through a cell disrupter (Constant Systems) two times at 1.8 kbar. After removal of unlysed cells by centrifugation (4000 x $g$ for 15 min), the membrane fraction was prepared by ultracentrifugation at 100,000 x $g$ for one hour, resuspended in lysis buffer and protein was solubilized by addition of 0.5% LMNG (Anatrace) for 1 h. The lysate was supplemented with 50 mM imidazole and loaded onto a 5 mL HisTrap FF column (GE Healthcare), washed with 6 column volumes (CV) wash buffer (20 mM Tris, 200 mM NaCl, 50 mM imidazole, 1 mM DTT, 0.005% LMNG, pH 7.5) and eluted with a gradient of 50 mM–1 M imidazole (20 mM Tris, 200 mM NaCl, 1 M imidazole, 1 mM DTT, 0.005% LMNG, pH 7.5) over 6 CV. Fractions containing FHL (70−300 mM imidazole) were pooled and concentrated to 2–4 mg/mL, and subsequently injected onto a Superdex 200 Increase 5/150 GL (GE Healthcare) size exclusion column pre-equilibrated with SEC buffer (20 mM Tris, 200 mM NaCl, 0.005% LMNG, 1 mM DTT). The SEC step revealed peaks due to the full FHL complex and the dissociated soluble arm (Supplementary Fig. 1a).

Anaerobic isolation of FHL was performed in a similar fashion, with the following modifications. Purification was conducted in an anaerobic chamber (<1 ppm $O_2$; Coy Laboratory Products) in an atmosphere of 3-4% hydrogen in nitrogen. All buffers were made anaerobic by extensive degassing and subsequent equilibration within the anaerobic chamber (>24 h). Cells were lysed on ice with an ultrasonicator (Bandelin) running 6 cycles of 1 min at 200 W. In contrast to the aerobic procedure, the reduced FHL complex could be eluted as a monodisperse peak from the size exclusion column, which only showed small amounts of a complex lacking the membrane subunits HycC and HycD (Supplementary Fig. 2a). FdhF was detached from the complex during purification under reducing conditions and only traces of an -80 kDa band could be identified by SDS-PAGE (Supplementary Fig. 2a). Repeated efforts to obtain a complete, anaerobically purified complex were unsuccessful. The cause for this is unclear, but it is worth noting that the difference cannot be due to differential expression, as aliquots of the same cells were used for both purification protocols. Small-scale aerobic purifications of cells lysed by sonication also allowed isolation of complex containing FdhF, while anaerobic purification following cell lysis by bead-beating or B-PER (Thermo Fisher Scientific) resulted in complex lacking FdhF, suggesting that the cell lysis method is not a deciding factor in copurification of the FdhF subunit.

## Cryo-EM sample preparation and data acquisition

Purified FHL was concentrated to 1–2 mg/mL using an Amicon centrifugal concentrator (100-kDa cutoff; Merck) and 3 μL of protein solution was applied to freshly glow-discharged C-flat 1.2/1.3 or 2/1 holey carbon grids (Protochips), automatically blotted and plunge frozen in liquid ethane using a Vitrobot Mark IV (Thermo Fisher Scientific) at 4 °C with 100% humidity. Cryo-EM grids of anaerobically purified FHL were prepared similarly within the anaerobic chamber except that 0.5x CMC fluorinated fos-choline-8 (Anatrace) was added immediately before the sample was applied to the grid. This resulted in a more even particle distribution and orientation within the ice (Supplementary Fig. 2b).

For the aerobically prepared sample a dataset of 1,207 dose-fractionated movies with 40 frames were recorded automatically with EPU (Thermo Fisher Scientific) on an FEI Titan Krios G2 transmission electron microscope operating at 300 kV equipped with a Bioquantum

energy filter and K2 direct electron detector (Gatan) at a pixel size of 0.828 Å operated in counting mode. The total dose was -72 e$^−$/Å$^2$ and the defocus range was set from −2.2 to −1.6 μm.

Cryo-EM data of anaerobically prepared FHL were acquired automatically on an FEI Titan Krios G2 transmission electron microscope operating at 300 kV equipped with a Bioquantum energy filter and K3 direct electron detector (Gatan) at a pixel size of 0.831 Å in counting mode. 7,338 dose-fractionated movies of anaerobically purified FHL were collected using fast acquisition by aberration-free image shift (AFIS) in EPU. Movies contained 61 frames at a total dose of -61 e$^−$/Å$^2$ and a defocus range from −2.5 to −1.0 μm.

## Image processing

Dose-fractionated movies of aerobically prepared FHL were subjected to motion correction and dose-weighting using MotionCor2[70] and the per-micrograph contrast transfer function (CTF) was determined by CTFFIND4.1[71] (major processing steps are visualized in Supplementary Fig. 1c). Initially, particles were picked manually from ten micrographs in order to train a model for neural-network picking in crYOLO[72], particle coordinates were imported into RELION-3.0[73] and 202,046 particles were extracted with a box size of 480 by 480 pixels. 2D classification was performed in cryoSPARC[74], 150,117 particles were selected and subjected to homogenous refinement using a de novo reconstruction generated from the same dataset. Particle coordinates were converted into a star-file using pyem[75] and reimported to RELION for 3D classification. The predominant 3D class (91,127 particles) was selected and refined to a resolution of 3.86 Å. Two rounds of CTF refinement and Bayesian polishing with intermediate 3D refinements yielded a consensus map with a global resolution of 3.40 Å. Focused refinements with masks covering individual parts of the complex gave resolutions of 3.06 Å (FdhF), 2.99 Å (soluble arm without FdhF) and 3.43 Å (membrane arm) respectively. An initial model was built using the globally refined maps, and this model was used as input with the focussed maps and consensus map to generate a composite map using phenix.combine_focused_maps[76] that was used for further model building and refinement (Supplementary Fig. 1d).

For the dataset of anaerobically purified FHL, initial image processing steps were performed in RELION-3.1 (major processing steps are visualized in Supplementary Fig. 2c). Movies were sorted into optics groups according to their EPU AFIS metadata[77] before motion correction and CTF estimation. Particles were initially picked on a micrograph subset with RELION's AutoPick implementation; positive picks based on appearance of 2D class averages were selected and used to train a particle-detection model with Topaz[78]. The Topaz model was used for particle picking from all micrographs, yielding 686,602 particles, which were initially extracted with a box size of 480 by 480 pixels and 4-fold down-sampling. Initial 2D classes were selected based on interpretable features and used for de novo initial model generation. In a first, coarse 3D classification all particles were aligned to the initial model, lowpass filtered to 60 Å. The major 3D class contained 565,559 particles that were subjected to another round of 3D classification using the same parameters. The major class, containing 300,443 particles, was selected and extracted with a box size of 384 by 384 pixels and 2-fold down-sampling. 3D auto refinement gave a map with 3.42 Å resolution. CTF refinement was performed on unbinned particles (box size 300 by 300 pixels) to correct for aberrations, anisotropic magnification and per-particle defocus, with each AFIS position being refined as an independent optics group[79]. Subsequent 3D refinement improved the resolution to 3.14 Å. Bayesian polishing was performed from trained parameters. Polished particles were imported to cryoSPARC and subjected to non-uniform refinement. All processing to this point was carried out with a pixel size of 0.837 Å/px. The half-maps were combined and the resulting map sharpened using relion_postprocess with the mask from the non-

uniform refinement and an updated calibrated pixel size of 0.831 Å/px yielding a final reconstruction with an average resolution of 2.62 Å.

## Model building and refinement

Templates for all subunits, except FdhF (PDB 1FDO[29]), were generated from homologous structures using the SWISS-MODEL server[80]. The cryo-EM structure of *P. furiosus* MBH (PDB 6CFW[27]) chain M (coverage 179-537) served as a template for the HycE C-terminus, chain A (coverage 11-488) and D (coverage 469-606) for HycC N- and C-terminus, chain E (coverage 9-303) for HycD and chain K (coverage 25-168) for HycG. The complex I structures PDB 6GCS[81] chain G (coverage 8-153) and PDB 5LDX[82] chain I (coverage 14-132) were used as templates for the HycE N-terminus and HycF, respectively. For HycB a template from the arsenate respiratory reductase PDB 6CZ7[83] chain B (coverage 1-171) was used.

Homology models were initially rigid body fitted into the cryo-EM density of aerobically isolated FHL, followed by manual building and refinement in Coot[84], from sequence information and secondary structure prediction[85]. The final models contained 2690 residues (aerobic preparation) and 2,026 residues (anaerobic preparation without FdhF). Cofactors were built in Coot and respective restraints were added using ReadySet! within PHENIX[76]. Phospholipids were added to the model obtained from anaerobically isolated sample, where the density allowed identification (1 CDL, 2 PE, 1 PG, 1 LMNG). In addition, water molecules were modeled in well-defined densities with reasonable hydrogen-bonding distances to the polypeptide. The models were refined against the map using phenix.real_space_refine[76] and validated by MolProbity[86] (for model statistics see Supplementary Table 1). All figures of models and maps were prepared with ChimeraX[87].

## Hydrogen-uptake assay

Hydrogenase activity was assessed by hydrogen-dependent benzyl viologen reduction[88]. Anaerobic gas-tight rubber-stoppered cuvettes were filled with anaerobic buffer (20 mM Tris, 200 mM NaCl, 1 mM DTT, pH 7.5) supplemented with 4 mM benzyl viologen (Sigma-Aldrich) and the headspace was saturated with $H_2$. Enzyme was added and benzyl viologen reduction was monitored photometrically at a wavelength of 598 nm. The rate of benzyl viologen reduction was calculated from the initial slopes assuming a molar extinction coefficient of $7400\ M^{-1}\ cm^{-1}$. This rate was converted to $mol(H_2) \cdot mol(FHL)^{-1} \cdot s^{-1}$ by dividing by 2.

## Sequence alignments

Alignments were generated with ClustalW[89] and visualized with the ESPript server[90].

# Data availability

Atomic models and maps for anaerobically isolated FHL were deposited in the Protein Data Bank with accession code 7Z0S and Electron Microscopy Data Bank with accession code EMD-14429; for aerobically isolated FHL models and maps are available as 7Z0T and EMD-14430. Composite map EMD-14430 is derived from focused refined maps accessible under EMD-14431, EMD-14432, EMD-14433, and EMD-14434 (see Supplementary Tables 1–2 for details). Other data are available from the corresponding author upon request. Previously published atomic models used as homology templates for atomic model building and within figures are available in the Protein Data Bank with access codes 4HEA, 1FDO, 2IV2, 4U9H, 7O71, 6CFW, 6GCS, 5LDX, and 6CZ7. Source data are provided with this paper.

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

## Acknowledgements

We thank the Central Electron Microscopy Facility of the Max Planck Institute of Biophysics for providing cryo-EM infrastructure and technical support, and Werner Kühlbrandt and the department of Structural Biology for support and helpful discussions. We are grateful to Frank Sargent for providing *E. coli* strain MG059e1 and for helpful discussions. We thank Stella Vitt and Rita Zimmermann for sharing their valuable expertise in anaerobic work. We thank Volker Zickermann and Janet Vonck for critical reading and comments on the manuscript. This work was funded by the Max Planck Society (to B.J.M.) and the Deutsche Forschungsgemeinschaft (DFG, German Research Foundation)—Project-ID 450648163—CRC 1507/1 2022 to B.J.M.

## Author contributions

R.S. purified protein and carried out activity assays, cryo-EM sample preparation, data acquisition, data processing, data analysis and interpretation, atomic model building, preparation of figures, manuscript preparation and revision. G.H. carried out anaerobic fermentation and anaerobic cell harvesting, and revised the manuscript. J.H. provided fermentation infrastructure and revised the manuscript. B.J.M. supervised the project and carried out cryo-EM sample preparation, data acquisition, data analysis and interpretation, manuscript preparation and revision.

## Funding

## Competing interests

The authors declare no competing interests.
