## [Peer Review File · Nature Communications]

**REVIEWER COMMENTS**

**Reviewer #1 (Remarks to the Author):**

**The manuscript by Steinhilper et al. describes the first cryo-EM structure of the formate**
**hydrogenlyase complex from E. coli. The data presented in this manuscript are important for a**
**number of reasons, not least because of the structural and functional similarities (and differences)**
**with complex I of the respiratory chain. The comparatively high resolution of the structures of the**
**complex that is presented helps explain many earlier observations pertaining to the instability of**
**the complex and why this complex remained refractory to biochemical characterization for so**
**long. While the manuscript is generally well presented, there are a few issues that the authors**
**need to address.**

**1. The Introduction ends rather abruptly and requires minimally a concluding sentence. The reader**
**is left wondering what the overall aim of the study is. Simply put, what question is being asked**
**and how will the cryo-EM structure provide an answer to this question?**

**2. Care should be taken in scientific terminology and descriptions. For example, on line 83 the**
**authors write ‘Size exclusion chromatography (SEC) of the aerobically isolated complex elutes as**
**two peaks...’ A verb is missing because this scans to indicate that the SEC elutes as two peaks and**
**not the complex.**

**3. This reviewer has a problem with the rather speculative proposal on lines 134-136 and 194-202**
**that presume ‘reversible’ loss FdhF from the complex has some kind of regulatory function under**
**strongly reducing conditions to prevent formate generation. The authors should provide direct**
**evidence for this proposal. This reads a bit like ‘science fiction’ to try to explain the newly**
**identified metal site. If concrete evidence (in vivo, not in vitro) supporting this hypothesis cannot**
**be presented, the corresponding text should be removed from the manuscript, particularly in view**
**of the fact that the Sargent group has clearly demonstrated that under high pressure and high gas**
**concentrations, H₂ and CO₂ are converted very efficiently to formate by whole cells using the FHL**
**complex (see reference 10). If the authors’ proposal were correct, then formate production, as**
**demonstrated by Roger et al., should not work, or would function at best very inefficiently.**

**4. Line 167. Are the authors really sure that the three Fdhs in E. coli can be legitimately described**
**in biochemical terms as isoenzymes?**

**5. Line 192, ‘close proximity’ is tautological; one or other would suffice.**

**6. An intriguing aspect of FHL is that, while each activity (Fdh-H and Hyd-3) can be measured**
**separately using redox dyes in extracts derived from a mutant lacking the HycD membrane**
**components, H₂ and CO₂ production from formate cannot be demonstrated either in vitro or in**
**vivo for such a mutant. This has been nicely demonstrated and dissected genetically and**
**biochemically by Pinske and Sargent in 2016 (reference 20). The structure of HycD shown in the**
**current manuscript, and how it links to the ‘soluble’ part of the complex, provides a likely**
**biochemical/structural explanation for the requirement of an intact complex for H₂ and CO₂**
**production. The authors should discuss their structural data, in particular, the hydrophilic E-**
**channel, in light of the results described in reference 20.**

**Reviewer #2 (Remarks to the Author):**

**The manuscript by Steinhilper and colleagues describes the cryo EM structure of formate hydrogen**
**lyase from E. coli, the membrane protein complex responsible for production of CO₂ and H₂ from**

formate under anaerobic growth conditions. The structural characterization of this complex is of
great ecological and economic interest, as the enzyme represents a potential source for renewable
H₂ production and/or reduction of carbon dioxide to formic acid.

The authors present structures after aerobic and anaerobic isolation of the complex, reaching a
resolution up to 2.6Å. Both the isolation of the complex as well as the structural work are
remarkably well done and represent an important scientific achievement. The information gained
from this investigation is undoubtedly of high interest to the scientific community and contributes
to our understanding of enzymes catalyzing coupled redox reactions. Nevertheless, the authors
should clarify some issues before publication.

Besides the structure itself, the most intriguing result is that isolation of the complex under
anaerobic conditions leads to a loss of FdhF, which is explained by a destabilization of the
FdhF/HycB interface through side-chain rotation of Arg122. Albeit this is an interesting hypothesis,
it is not very well substantiated with experimental data. The authors should provide some
evidence to support this hypothesis such as a mutational study, re-binding experiments after H₂
removal or adjustment of the redox potential during purification. Also, it is very difficult to
understand where in the structure the Arg122 and the metal ion are found. An overview of the
region of interest and the position of the Arg122 containing loop in the enzyme, perhaps as an
enlarged section of Figure 1b should be depicted in Figure S5. The side chain rotation is inferred
from the EM map of the anaerobically purified complex, but it is not easy to make out the
densities in Figure S5. A comparison of the densities of the two structures (aerobically,
anaerobically) in this region would make it easier to follow the reasoning.

Another interesting hypothesis in this manuscript describes that the enhanced hydrogen
production by Ec Hyd-3 is due to either increased flexibility of loop beta1-beta2 or a serine residue
in the active site. An investigation of the effects of an S283D mutation on hydrogen production
(even in vivo) would support this statement with experimental evidence.

It is incomprehensible why Figure 4 appears in the main part, while Figure S3 and Figure S5, which
describe important findings of the manuscript, are placed in the supplements. In my opinion, this
prioritization should be reversed.

minor comments:

1) some parts of the manuscript would be easier to understand if nomenclature of FeS clusters
introduced in Figure 1 were used throughout; e.g.

- line 99: and the proximal iron-sulfur cluster G1 in HycG

- line 173f: loop region between the B3 (third) and B4 (fourth) 4Fe4S-cluster-ligating-cysteine...

- line 189: from the nearest 4Fe4S cluster (B3) of HycB

- line 190: from the nearest 4Fe4S cluster (F1?) in HycF

2) line 240 should read: ...HydB (Fig 3d).

Reviewer #3 (Remarks to the Author):

In this paper, Steinhilper and co-workers use cryo-EM techniques to determine the structure of
formate hydrogenlyase (FHL-1) from Escherichia coli. They use a genetically engineered strain
(rather than a plasmid-based system) grown at enormous scale, and then highly skilled anaerobic
purification techniques. What they end up with is one of the most exciting advances in bacterial

physiology in recent times. FHL-1 is an incredible enzyme with links in to fundamental
bioenergetics but also has huge biotechnological importance with its use of H₂ and CO₂ as
substrates. Finally having a real structure of this enzyme is a cause for celebration.

Two structures are presented. One purified under aerobic conditions, which is essentially
'complete' and has the FdhF formate dehydrogenase in position (but the metal cofactors are
understandably damaged). A second structure was determined from protein purified under
anaerobic (including H₂) conditions and this one lacks the formate dehydrogenase. One exciting
new discovery is a small metal-containing domain that seems to govern an ionic interaction
between FdhF and HycF.

Overall, the paper is a wonderful read with an understandable focus on similarities with Complex
I.

POINTS OF CLARIFICATION

1. FdhF attachment

The observation that FdhF is not attached to the Hyd3 component under reducing conditions
(presumably reduced as it was isolated in 4% H₂) is an important observation, however the
speculation that this is to prevent full reversal of the FHL reaction seems to be disproved by the
published data. Hydrogen-Dependent CO₂-Reduction by FHL has been reported a few times.
Instead, it would be worth thinking again about this hypothesis and really consider accepting that
FdhF may always be a transient interactor with HycB/F, and that it is the oxidised form that
fortuitously traps FdhF in its attachment mode. Transient interactions are very common in
electron transport chains. TMAO reductase, for example, is only a transient interactor with its
electron donor TorC, or another good example would be cytochrome c peroxidase, which is a
transient interactor with cytochrome c550. In all of these examples there will be an on-rate (when
the reduced donor interacts with the oxidised recipient) and an off-rate (when the now oxidised
donor leaves the now reduced recipient). Maintaining a fast dissociation rate is commonly
achieved by having a relatively weak association governed by hydrophobic interactions (as is
apparently the case here with FHL-1). And achieving correct orientation of the donor protein with
the recipient is commonly achieved by having a limited number of ionic interactions (as is also the
case here). There is a large body of work looking at transient fast electron transfer between
proteins and it would really be worth discussing this in the manuscript (e.g. DOI:
10.1038/nsb1195-975). Also, bear in mind that FdhF 'moonlights' in other biochemical pathways
(DOI: 10.1128/JB.00573-18) so it could not be permanently bound to Hyd3, and that FdhF-
homologues exist in *E. coli* (e.g. YdeP) that, under some as yet unreported conditions, may wish to
pass electrons to Hyd-3.

It therefore makes some sense to this reviewer that a full reduced recipient should not have the
oxidised donor attached. It also makes sense that an oxidised recipient should have some donor
attached.

And one final thought on this issue - the formate dehydrogenase and the hydrogenase will have
different reaction rates. Presumably the Hyd-3 H₂ evolution rate is much faster than the formate
oxidation rate of FdhF?

**2. E-channel hypothesis**

**The E-channel of proton movement in to the active site of the hydrogenase is a fascinating aspect**
**of this paper worth clarifying here. Is the hypothesis that protons destined to be reduced to H₂ are**
**plucked from the cell cytoplasm via HycC and HycD in the membrane and directed to the active**
**site? It's important to consider this because the enzyme in vivo is not active in the absence of HycC**
**or HycD nor when the ND1 loop residues are modified. Could the protons be coming instead from**
**the P-side of the membrane rather than the N-side?**

**3. Proton Pumping**

**Is there anything else to say on this other than it really looks like it should?**

Reviewer #1 (Remarks to the Author):

The manuscript by Steinhilper et al. describes the first cryo-EM structure of the formate hydrogenlyase complex from *E. coli*. The data presented in this manuscript are important for a number of reasons, not least because of the structural and functional similarities (and differences) with complex I of the respiratory chain. The comparatively high resolution of the structures of the complex that is presented helps explain many earlier observations pertaining to the instability of the complex and why this complex remained refractory to biochemical characterization for so long. While the manuscript is generally well presented, there are a few issues that the authors need to address.

1. The Introduction ends rather abruptly and requires minimally a concluding sentence. The reader is left wondering what the overall aim of the study is. Simply put, what question is being asked and how will the cryo-EM structure provide an answer to this question?

We have added the following text on Line72ff:

Until now a structural understanding of FHL has been lacking, impeding further insight into its role in bacterial bioenergetics, its bias towards hydrogen production and its links to other members of the complex I superfamily. In this report, we present full atomic models of the complex in the oxidized and reduced states, providing important insight into the function and regulation of FHL and related complexes.

2. Care should be taken in scientific terminology and descriptions. For example, on line 83 the authors write 'Size exclusion chromatography (SEC) of the aerobically isolated complex elutes as two peaks...' A verb is missing because this scans to indicate that the SEC elutes as two peaks and not the complex.

We have corrected this error (Line 89ff) and taken care to avoid such errors throughout the text:

The size exclusion chromatography (SEC) elution profile of the aerobically isolated complex shows two major peaks, the slower-eluting of which corresponds to a soluble complex lacking the membrane-associated subunits HycC and HycD (Fig. S1a). In contrast, the SEC profile of anaerobically purified protein shows a single predominant peak, with a much smaller peak corresponding to the complex without HycC/HycD (Fig. S2a).

3. This reviewer has a problem with the rather speculative proposal on lines 134-136 and 194-202 that presume 'reversible' loss FdhF from the complex has some kind of regulatory function under strongly reducing conditions to prevent formate generation. The authors should provide direct evidence for this proposal. This reads a bit like 'science fiction' to try to explain the newly identified metal site. If concrete evidence (in vivo, not in vitro) supporting this hypothesis cannot be presented, the corresponding text should be removed from the manuscript, particularly in view of the fact that the Sargent group has clearly demonstrated that under high pressure and high gas concentrations, H₂ and CO₂ are converted very efficiently to formate by whole cells using the FHL complex (see reference 10). If the authors' proposal were correct, then formate production, as demonstrated by Roger et al., should not work, or would function at best very inefficiently.

*We have taken this point onboard and have modified the text accordingly. We did not intend to suggest that the HDCR reaction is impossible - clearly it does occur. However, our data strongly support the idea that the reduced Hyc complex has a lower affinity for FdhF than does the oxidized Hyc complex. Since the HDCR reaction requires electrons to be transferred from (reduced) Hyc to FdhF, we would expect this change in affinity to lower the rate of the HDCR. On the other hand, we would not expect any effect on the efficiency as reported by Roger et. al, 2018, which is defined in that manuscript as the ratio of CO₂ taken up to formate produced. For this reason, we do not see that our data are inconsistent with the findings of the Sargent group on the HDCR. We also note that the strains MR40 and MR60 that were used for many of the experiments in the Roger 2021 paper encode an FdhF-HycB fusion protein rather than the wild-type complex. We have removed the reference in the abstract to 'preventing reverse activity *in vivo*' and have revised other sections of the text as below:*

Line 133ff:

Our data suggest that FdhF is more strongly associated with the Hyc subunits under oxidizing than under reducing conditions. This may be related to a regulatory process, although we cannot conclusively rule out

the possibility that other difficult-to-control differences between aerobic and anaerobic protocols could lead to the differing observations.

Line 196ff

We suggest that a redox-active iron in this binding site could play a role in regulating affinity of the complex for FdhF as reduction of the iron would decrease the strength of charge-charge interaction between the two sites. Such a mechanism would be expected to lower the rate of hydrogen-dependent CO₂ reduction by the complex, since this reaction requires interaction of a reduced Hyc complex with FdhF.

4. Line 167. Are the authors really sure that the three Fdhs in E. coli can be legitimately described in biochemical terms as isoenzymes?

We have changed 'isoenzyme' to 'homologue' (line 169)

5. Line 192, 'close proximity' is tautological; one or other would suffice.

The manuscript now reads 'Arg122 of HycF, which is near to the ion...' (line 201)

6. An intriguing aspect of FHL is that, while each activity (Fdh-H and Hyd-3) can be measured separately using redox dyes in extracts derived from a mutant lacking the HycCD membrane components, H₂ and CO₂ production from formate cannot be demonstrated either in vitro or in vivo for such a mutant. This has been nicely demonstrated and dissected genetically and biochemically by Pinske and Sargent in 2016 (reference 20). The structure of HycD shown in the current manuscript, and how it links to the 'soluble' part of the complex, provides a likely biochemical/structural explanation for the requirement of an intact complex for H₂ and CO₂ production. The authors should discuss their structural data, in particular, the hydrophilic E-channel, in light of the results described in reference 20.

We thank the reviewer for this helpful comment and have added the following text to better integrate our structural data with previous mutational studies:

Line 334ff:

Site-directed mutagenesis of HycD shows that substitutions of residues along this pathway, reaching as far as E138 at the interface of HycD/HycC (Fig. 6c), strongly impact hydrogen production in vivo¹⁰.

Line 349ff:

This observation is consistent with a previous mutational study showing that HycC mutants E391A and D354A reduced hydrogen production by 45%, whereas E135A, H222A, K239A, T292A, H282A and K336A mutations did not cause a significant decrease in hydrogen production in vivo¹⁰.

Reviewer #2 (Remarks to the Author):

The manuscript by Steinhilper and colleagues describes the cryo EM structure of formate hydrogen lyase from E. coli, the membrane protein complex responsible for production of CO₂ and H₂ from formate under anaerobic growth conditions. The structural characterization of this complex is of great ecological and economic interest, as the enzyme represents a potential source for renewable H₂ production and/or reduction of carbon dioxide to formic acid.

The authors present structures after aerobic and anaerobic isolation of the complex, reaching a resolution up to 2.6Å. Both the isolation of the complex as well as the structural work are remarkably well done and represent an important scientific achievement. The information gained from this investigation is undoubtedly of high interest to the scientific community and contributes to our understanding of enzymes catalyzing coupled redox reactions. Nevertheless, the authors should clarify some issues before publication.

Besides the structure itself, the most intriguing result is that isolation of the complex under anaerobic conditions leads to a loss of FdhF, which is explained by a destabilization of the FdhF/HycB interface through side-chain rotation of Arg122. Albeit this is an interesting hypothesis, it is not very well substantiated with experimental data. The authors should provide some evidence to support this hypothesis such as a mutational study, re-binding experiments after H₂ removal or adjustment of the redox potential during purification. Also, it is very difficult to understand where in the structure the Arg122 and the metal ion are found. An overview of the region of interest and the position of the Arg122 containing loop in the enzyme, perhaps as an enlarged section of Figure 1b should be depicted in Figure S5. The side chain rotation is inferred from the EM map of the anaerobically purified complex, but it is not easy to make out the densities in Figure S5. A comparison of the densities of the two structures (aerobically, anaerobically) in this region would make it easier to follow the reasoning.

We agree with the reviewer that the proposed redox-dependent interaction is a hypothesis that requires testing. Upon re-reading the section, we feel our ambiguous wording may have led to a slight misunderstanding: although we feel that the observation of the Arg-122 conformation is worth mention and could play a role, the general

proposal is that an oxidized metal ion, by virtue of its higher positive charge, will contribute more strongly to interaction with the negatively-charged patch on FdhF than will a reduced metal ion. We have modified the text, in particular by changing the order of statements, to more clearly convey the hypothesis. Lines 196-198 now read:

We suggest that a redox-active iron in this binding site could play a role in regulating affinity of the complex for FdhF as reduction of the ion would decrease the strength of charge-charge interaction between the two sites.

We agree with the importance of following up on this observation with further studies but this is outside the scope of the present work. We have added a figure to indicate the location of the ion and a more detailed comparison of the density for Arg-122 to Fig. 4b.

Another interesting hypothesis in this manuscript describes that the enhanced hydrogen production by Ec Hyd-3 is due to either increased flexibility of loop beta1-beta2 or a serine residue in the active site. An investigation of the effects of an S283D mutation on hydrogen production (even in vivo) would support this statement with experimental evidence.

We agree with the reviewer that this would indeed be an interesting line of enquiry, but a proper treatment of this question is not within the scope of this paper.

It is incomprehensible why Figure 4 appears in the main part, while Figure S3 and Figure S5, which describe important findings of the manuscript, are placed in the supplements. In my opinion, this prioritization should be reversed.

We agree with the reviewer and have changed the figures accordingly.

minor comments:

7. some parts of the manuscript would be easier to understand if nomenclature of FeS clusters introduced in Figure 1 were used throughout; e.g.
 - line 99: and the proximal iron-sulfur cluster G1 in HycG
 - line 173f: loop region between the B3 (third) and B4 (fourth) 4Fe4S-cluster-ligating-cysteine...
 - line 189: from the nearest 4Fe4S cluster (B3) of HycB
 - line 190: from the nearest 4Fe4S cluster (F1?) in HycF

Done, thank you.

8. line 240 should read: ...HydB (Fig 3d).

Done, thank you.

Reviewer #3 (Remarks to the Author):

In this paper, Steinhilper and co-workers use cryo-EM techniques to determine the structure of formate hydrogenlyase (FHL-1) from Escherichia coli. They use a genetically engineered strain (rather than a plasmid-based system) grown at enormous scale, and then highly skilled anaerobic purification techniques. What they end up with is one of the most exciting advances in bacterial physiology in recent times. FHL-1 is an incredible enzyme with links in to fundamental bioenergetics but also has huge biotechnological importance with its use of H₂ and CO₂ as substrates. Finally having a real structure of this enzyme is a cause for celebration.

Two structures are presented. One purified under aerobic conditions, which is essentially 'complete' and has the FdhF formate dehydrogenase in position (but the metal cofactors are understandably damaged). A second structure was determined from protein purified under anaerobic (including H₂) conditions and this one lacks the formate dehydrogenase. One exciting new discovery is a small metal-containing domain that seems to govern an ionic interaction between FdhF and HycF.

Overall, the paper is a wonderful read with an understandable focus on similarities with Complex I.

POINTS OF CLARIFICATION

9. FdhF attachment

The observation that FdhF is not attached to the Hyd3 component under reducing conditions (presumably reduced as it was isolated in 4% H₂) is an important observation, however the speculation that this is to prevent full reversal of the FHL reaction seems to be disproved by the published data. Hydrogen-Dependent CO₂-Reduction by FHL has been reported a few times. Instead, it would be worth thinking again about this hypothesis and really consider accepting that FdhF may always be a transient

interactor with HycB/F, and that it is the oxidised form that fortuitously traps FdhF in its attachment mode. Transient interactions are very common in electron transport chains. TMAO reductase, for example, is only a transient interactor with its electron donor TorC, or another good example would be cytochrome c peroxidase, which is a transient interactor with cytochrome c550. In all of these examples there will be an on-rate (when the reduced donor interacts with the oxidised recipient) and an off-rate (when the now oxidised donor leaves the now reduced recipient). Maintaining a fast dissociation rate is commonly achieved by having a relatively weak association governed by hydrophobic interactions (as is apparently the case here with FHL-1). And achieving correct orientation of the donor protein with the recipient is commonly achieved by having a limited number of ionic interactions (as is also the case here). There is a large body of work looking at transient fast electron transfer between proteins and it would really be worth discussing this in the manuscript (e.g. DOI: 10.1038/nsb1195-975). Also, bear in mind that FdhF 'moonlights' in other biochemical pathways (DOI: 10.1128/JB.00573-18) so it could not be permanently bound to Hyd3, and that FdhF-homologues exist in *E. coli* (e.g. YdeP) that, under some as yet unreported conditions, may wish to pass electrons to Hyd-3.

It therefore makes some sense to this reviewer that a full reduced recipient should not have the oxidised donor attached. It also makes sense that an oxidised recipient should have some donor attached.

And one final thought on this issue - the formate dehydrogenase and the hydrogenase will have different reaction rates. Presumably the Hyd-3 H₂ evolution rate is much faster than the formate oxidation rate of FdhF?

If we have understood correctly, the reviewer is suggesting the possibility that FdhF interacts transiently with the complex, perhaps even dissociating on the timescale of an individual turnover cycle. It is difficult to evaluate this proposal on the basis of our data, except to say that in the oxidized state, the off-rate should be on the order of tens of minutes at least (the wash step on the Ni-NTA column takes ~30 min, so the off-rate shouldn't be higher than 1/15 min \approx 0.001s⁻¹). For a transient interaction in which oxidized FdhF is exchanged for reduced FdhF in each turnover cycle, the off rate would have to increase by at least five orders of magnitude between oxidized and reduced conditions to keep up with turnover at ~100s⁻¹. Whether FdhF has a stable but fragile connection to the rest of the complex, or dissociates and reassociates on the timescale of turnover, is a question that would require further investigation. Regardless of the timescale, the impact of this change in affinity would be to lower the rate of the HDCR reaction, since that reaction can only occur when FdhF interacts with reduced Hyc complex, see also our reply to Reviewer 1 (point 3). The existing evidence is clear that the HDCR reaction can and does still occur - the question is only at what rate.

We have modified the text to include the reviewer's insights: we have removed our reference to a 'stable but fragile' interaction of FdhF and added a comment related to interaction of FdhF with other partners in the cell (lines 137f). We have clarified the anticipated effect on the rate of HDCR (line 198-203).

10. E-channel hypothesis

The E-channel of proton movement in to the active site of the hydrogenase is a fascinating aspect of this paper worth clarifying here. Is the hypothesis that protons destined to be reduced to H₂ are plucked from the cell cytoplasm via HycC and HycD in the membrane and directed to the active site? It's important to consider this because the enzyme *in vivo* is not active in the absence of HycC or HycD nor when the ND1 loop residues are modified. Could the protons be coming instead from the P-side of the membrane rather than the N-side?

Thank you for this point. We have added further discussion relating the mutational study of HycC and D to our structure (lines 334-356), including:
'We did not observe a clear proton pathway connecting the E-channel to the P-side of the membrane' (line 355-356).

11. Proton Pumping

Is there anything else to say on this other than it really looks like it should?

We cannot confidently state whether FHL does or does not translocate protons across the membrane, but we are optimistic that the added discussion on proton pathways through HycCD and integration with published mutational data advances the discussion on this long-standing mystery.

**REVIEWERS' COMMENTS**

Reviewer #1 (Remarks to the Author):

The authors have addressed adequately all of the original comments made by this reviewer, and in
this reviewer's opinion, also the comments made by the other reviewers have been adequately
addressed.

This manuscript reports on a very important structure with wide-ranging evolutionary and
bioenergetic implications.

I have only a very few remaining and very trivial editorial issues.

1. Line 46. Remove the semi-colon and replace it with a comma.

2. Line 53. Change the beginning of sentence to "Studies using inverted vesicles derived from
anaerobically grown..."

3. Line 181. Presumably, the authors mean to say "...influence on electron transfer based upon
structural dynamics.?" Otherwise, this makes no sense.

4. Line 338. 'through'

5. Line 405. Do not confuse proteins and genes. The hycE gene harbors additional codons
encoding a His10-tag placed after the codon encoding Gly-83.

6. Please check the formatting of the reference titles for uniformity.

Reviewer #2 (Remarks to the Author):

The authors have addressed all of my concerns apart from the suggestion to provide experimental
evidence for the hypothesis that S283 might be responsible for the low degree of product inhibition
during H₂ production in Hyd-3. Albeit the authors decided not to substantiate (any of) their
hypotheses drawn from the structures with experimental data, I would recommend publication,
since the structures itself are of high value to the scientific community. The manuscript is written
in a way that all hypotheses can be identified as such.

Reviewer #3 (Remarks to the Author):

No further comments

Reviewer #1 (Remarks to the Author):

The authors have addressed adequately all of the original comments made by this reviewer, and in this reviewer's opinion, also the comments made by the other reviewers have been adequately addressed. This manuscript reports on a very important structure with wide-ranging evolutionary and bioenergetic implications.

I have only a very few remaining and very trivial editorial issues.

1. Line 46. Remove the semi-colon and replace it with a comma.

Done, thank you.

2. Line 53. Change the beginning of sentence to "Studies using inverted vesicles derived from anaerobically grown..."

Line 53 now reads:

Studies using inverted membrane vesicles derived from anaerobically grown E. coli...

3. Line 181. Presumably, the authors mean to say "...influence on electron transfer based upon structural dynamics."? Otherwise, this makes no sense.

Reference 42 presents molecular dynamics simulations that suggest an influence of cardiolipin on the overall structural dynamics of complex I and the subsequent influence on quinone access to the active site. We find no reference in that work to a specific impact on electron transfer.

We have rephrased the sentence to better convey this.

Line 183f now reads: '*... with a suggested influence upon global structural dynamics and quinone access to the active site*⁴².'

4. Line 338. 'through'

Done, thank you.

5. Line 405. Do not confuse proteins and genes. The hycE gene harbors additional codons encoding a His10-tag placed after the codon encoding Gly-83.

Done, thank you.

6. Please check the formatting of the reference titles for uniformity.

Done, thank you.

Reviewer #2 (Remarks to the Author):

The authors have addressed all of my concerns apart from the suggestion to provide experimental evidence for the hypothesis that S283 might be responsible for the low degree of product inhibition during H2 production in

Hyd-3. Albeit the authors decided not to substantiate (any of) their hypotheses drawn from the structures with experimental data, I would recommend publication, since the structures itself are of high value to the scientific community. The manuscript is written in a way that all hypotheses can be identified as such.

Reviewer #3 (Remarks to the Author):

No further comments